# MATR3’s Role beyond the Nuclear Matrix: From Gene Regulation to Its Implications in Amyotrophic Lateral Sclerosis and Other Diseases

**DOI:** 10.3390/cells13110980

**Published:** 2024-06-05

**Authors:** Jhune Rizsan Santos, Jeehye Park

**Affiliations:** 1Department of Molecular Genetics, University of Toronto, Toronto, ON M5S 1A1, Canada; jhunerizsan.santos@mail.utoronto.ca; 2Genetics and Genome Biology Program, Peter Gilgan Centre for Research and Learning, The Hospital for Sick Children, Toronto, ON M5G 0A4, Canada

**Keywords:** MATR3, nucleic acid binding protein, DNA-binding protein, RNA-binding protein, amyotrophic lateral sclerosis

## Abstract

Matrin-3 (MATR3) was initially discovered as a component of the nuclear matrix about thirty years ago. Since then, accumulating studies have provided evidence that MATR3 not only plays a structural role in the nucleus, but that it is also an active protein involved in regulating gene expression at multiple levels, including chromatin organization, DNA transcription, RNA metabolism, and protein translation in the nucleus and cytoplasm. Furthermore, MATR3 may play a critical role in various cellular processes, including DNA damage response, cell proliferation, differentiation, and survival. In addition to the revelation of its biological role, recent studies have reported MATR3’s involvement in the context of various diseases, including neurodegenerative and neurodevelopmental diseases, as well as cancer. Moreover, sequencing studies of patients revealed a handful of disease-associated mutations in *MATR3* linked to amyotrophic lateral sclerosis (ALS), which further elevated the gene’s importance as a topic of study. In this review, we synthesize the current knowledge regarding the diverse functions of MATR3 in DNA- and RNA-related processes, as well as its involvement in various diseases, with a particular emphasis on ALS.

## 1. Introduction

Matrin-3 (MATR3) was initially discovered as one of the major components of the inner nuclear matrix [1,2]. It has also been independently discovered as an A/T-rich DNA-binding nuclear scaffold protein initially named P130, and as a hypo-phosphorylated variant named P123, but was later renamed as MATR3 [3,4,5,6,7]. MATR3 is primarily localized to the nucleus, consistent with its primary sequence containing a bipartite nuclear localization signal (NLS) [7,8]. It also encodes a nuclear export signal (NES) [7] which may suggest that MATR3 may also play a role in the cytoplasm. In addition, MATR3 has two C2H2 zinc finger (ZnF) domains which allow for binding to DNA, and two RNA recognition motifs (RRM) which allow for binding to RNA [7] (Figure 1).

As MATR3 was initially described as a nuclear matrix protein, MATR3 may play a structural or scaffolding role. However, accumulating studies since its discovery have demonstrated its role in various molecular and cellular processes involving DNA and RNA, including chromatin organization [9,10,11], DNA transcription and repair [6,12,13,14], and RNA splicing [15,16,17,18,19] (Figure 1). Recent studies have investigated its biological function in various cell types, including induced pluripotent stem cells, neuronal stem cells, differentiated neurons, and muscle cells, expanding our knowledge of the roles that MATR3 plays in modulating cellular health and function [10,11,12,20,21,22,23]. In addition, there have been an increasing number of studies implicating MATR3 in a variety of diseases. Most notably, missense mutations in MATR3 have been linked to amyotrophic lateral sclerosis (ALS), a fatal neurodegenerative disease caused by the loss of motor neurons [24], as well as in neurodevelopmental diseases [16,25]. Furthermore, recent studies have revealed that MATR3 may play a critical role in other diseases, including cancer [26,27,28,29,30,31].

Here, we summarize the reported roles that MATR3 plays in both DNA- and RNA-associated processes. In addition, we discuss the role of MATR3 in the modulation of cellular health and function. Lastly, we touch upon the involvement of MATR3 in the context of disease, with a particular emphasis on ALS.

## 2. MATR3’s Role in DNA-Related Processes

Accumulating studies have shown that MATR3 plays a role in a variety of functions related to DNA (Figure 1). MATR3 contains two C2H2-type ZnF domains that work cooperatively to bind DNA. Deletion of one ZnF domain reduces DNA binding, while deletion of both ZnF domains completely abolishes this capability [7]. Early studies have demonstrated that MATR3 recognizes and binds to repetitive, adenine/thymine (A/T)-rich DNA fragments, and that the DNA binding ability of MATR3 depends on its serine/threonine phosphorylation and the degree of the DNA bending [3,4,5]. MATR3 has also been demonstrated to bind to DNA fragments corresponding to the A/T-rich scaffold or matrix attachment region (S/MAR) [6], a chromosomal region that anchors chromatin to the nuclear matrix [32]. Additionally, MATR3 has been shown to associate with other S/MAR binding proteins, such as scaffold attachment factor A (SAFA) and scaffold attachment factor B (SAFB) [33]. Furthermore, immunostaining studies showed that MATR3 appears as a higher-order network-like structure that diffusely labels the nucleoplasm [9,33,34]. These studies provided clues that MATR3 may play a role in organizing chromatin structure. 

MATR3’s role in chromatin organization is further supported by protein–protein interaction studies showing that MATR3 interacts with a variety of chromatin remodelling factors as well as chromatin architectural proteins, such as CCCTC-binding factor (CTCF) and members of the cohesin complex [10,33,35] (Table 1). Moreover, loss of MATR3 led to a global reorganization of chromatin architecture in both mouse erythroleukemia (MEL) and mouse embryonic stem (mES) cells [10]. Furthermore, MATR3 loss in MEL cells decreased the chromatin occupancy of CTCF and the cohesin complex, which impaired proper chromatin interaction and expression of CTCF/cohesin-regulated genes, suggesting that MATR3 stabilizes the binding of CTCF/cohesin to chromatin, thereby maintaining the chromatin structure [10]. MATR3 loss in a mouse myoblast cell line similarly resulted in alterations to chromatin accessibility, chromatin loop domain interactions, occupancy of the CTCF and cohesin complex, and Yin Yang 1 (YY1)-mediated enhancer-promoter loop formation [11]. Another recent study has also proposed a role for MATR3 in the 3D organization of chromatin through association with antisense LINE1 [9]. X-chromosome inactivation (XCI) is an example of an extreme modification of chromatin composition/organization. Curiously, MATR3 has been recently linked to the maintenance of the inactive X-chromosome (X_i_) [36,37]. Specifically, MATR3, in conjunction with other RBPs, has been shown to interact with *Xist*, a long noncoding RNA that mediates XCI. Loss of this interaction led to *Xist* dispersal and reactivation of the genes on the X_i_. Taken together, it is interesting to note that despite its ability to bind DNA, MATR3’s role in chromatin organization does not seem to rely on its DNA-binding function but more on its ability to interact with other proteins and/or with RNA. More studies are needed to fully elucidate if MATR3’s DNA-binding capability is indeed dispensable in its ability to modify chromatin organization. 

Aside from its association with S/MAR [6,7], a chromosomal region that is implicated in transcription regulation [38], and its proximity to functional genomic areas associated with transcription [34], MATR3 has also been proposed to play a role in transcription regulation. Notably, an early work on MATR3 function provided evidence that MATR3 is involved in transcription [6]. Later studies refined the role of MATR3 in transcription and demonstrated that MATR3 binds to enhancer regions, in part due to its interaction with the transcription factor PIT1, and regulates the expression of PIT1-target genes [13]. MATR3 also has been shown to bind to the promoter regions of *OCT4* and *YTHDF1*, which are genes involved in the maintenance of stem cell pluripotency, as well as to the promoter of other genes involved in embryonic development and stemness [12]. Moreover, MATR3 has also been shown to interact with RNA Polymerase II [39,40] which adds another layer to MATR3’s role in transcription regulation. 

Another process that MATR3 has been implicated in is the DNA damage response (DDR), a complex network of signaling pathways that senses and responds to DNA damage. Proteomic interaction studies have shown that MATR3 interacts with proteins associated with the DDR or DNA repair, including the DNA-PK holoenzyme subunits Ku70 and Ku80, H2AX, RAD50, and RUVBL1/2 [33,39,41,42] (Table 1). Moreover, MATR3 has been found to be phosphorylated by DDR-associated kinases ATM [20,41,43] and Chk1 [44], and it has been found that the levels of ATM-phosphorylated MATR3 increased after DNA damage [41]. MATR3 has also been shown to interact with and modify the retention times of SFPQ and NONO [41], which have been previously implicated in DDR [45]. Another possible contributory role of MATR3 in the DDR is its ability to modulate the levels of other factors that may also play a role in DDR. Indeed, MATR3 depletion resulted in a decrease in the mRNA and protein levels of RAD51, a key player in DNA repair via homologous recombination (HR) [14]. More strikingly, MATR3 depletion led to a decrease in the formation of DNA damage-induced RAD51 foci and negatively impacted DNA repair by HR [14]. MATR3 has also been shown to regulate the levels of RAD17 and UHRF2 [16,46], which have both been shown to have roles in DDR [47,48,49,50].

**Table 1 cells-13-00980-t001:** MATR3 interacts with noncoding RNA as well as a variety of proteins involved in chromatin organization, nucleic acid metabolism, and protein translation.

Interactor	Ref	Interactor	Ref	Interactor	Ref
AGO1	[51]	RPL10A	[12,52]	RPS12	[12]
AGO2	[51]	RPL11	[12]	RPS13	[52,53]
ALYREF	[53]	RPL13/A	[52]	RPS14	[12]
BAZ1A	[33]	RPL14	[52]	RPS15A	[33]
CHD3	[33]	RPL15	[52]	RPS16	[12]
CTCF	[10]	RPL17	[12]	RPS18	[12,53]
DDX17	[54]	RPL18	[12,52]	RPS2	[52]
DUX4	[55]	RPL18A	[12,33]	RPS23	[12]
eEF1a1	[12]	RPL19	[12]	RPS27	[12]
EFTU	[12]	RPL22	[12]	RPS3	[12]
EIF3C	[52]	RPL23	[12]	RPS3A	[12,52]
EIF3CL	[12]	RPL23A	[12]	RPS4X	[12,52]
EIF3D	[12]	RPL26L1	[52]	RPS6	[52]
EIF3F	[12]	RPL27	[12,52,53]	RPS7	[12]
EIF4A1	[12]	RPL28	[12,52]	RPS8	[12]
EIF4A3	[12]	RPL3	[52]	RPS9	[12,52]
ESCO2	[52,53]	RPL30	[12]	RRBP1	[52]
EXOSC3	[54]	RPL32	[12]	RRP12	[53]
GARS	[12]	RPL36	[12]	RSL1D1	[52]
H2AX	[42]	RPL38	[12]	RUVBL1/2	[39]
Ku70, Ku80	[41]	RPL4	[52]	SAFB	[33,52]
*Neat1*	[21]	RPL5	[12]	SARNP	[53]
NONO	[12,41]	RPL6	[12,52]	SFPQ	[39,41,53]
*PINCR*	[56]	RPL7	[12,52]	SMARCA4	[33]
PIT1	[13]	RPL7A	[52]	SMC3	[10]
POLR2A	[40]	RPL8	[52]	SYDC	[12]
*pre-miR-138-2*	[57]	RPL9	[12]	SYIC	[12]
PTBP	[12,15,39,53]	RPLP0	[53]	SYLC	[12]
RAD21	[10]	RPLP2	[12]	*Xist*	[36,37]
RAD50	[39]	RPS11	[52,53]	ZAP (ZC3HAV1)	[52,54]
RPL10	[12,33,52]				

## 3. MATR3’s Role in RNA-Related Processes

MATR3 contains two RRM domains that can independently bind RNA [7], and the solution structures of both RRM domains have been previously resolved [58]. Deletion of both RRMs abolishes the RNA-binding capability of MATR3 [7]. Indeed, accumulating studies have demonstrated that MATR3 plays a role in RNA-related processes (Figure 1). Specifically, MATR3 has been shown to bind to and regulate mRNA stability [12,14,46,59]. MATR3 also plays a role in the nuclear retention of hyper-edited RNA [60], as well as in mRNA export through its interaction with the Transcription and Export (TREX) complex [53]. In addition, MATR3 has also been shown to bind to critical myogenic transcripts and regulate their RNA processing [21]. 

MATR3 has also been shown to interact with noncoding RNA, such as long noncoding RNA (lncRNA) and micro-RNA (miRNA). For instance, MATR3 binds to the lncRNA *Neat1,* and depletion of MATR3 increases *Neat1* levels by increasing its stability [21], suggesting a possible role of MATR3 in regulating *Neat1* lncRNA decay. Another lncRNA that has been known to bind with MATR3 is *PINCR*, which is strongly induced by DNA damage and regulated by p53 [56]. MATR3 has been shown to be important for the localization of the MATR3-*PINCR*-p53 complex to the enhancers of select p53/*PINCR*-regulated genes and plays a key role in modulating the expression of these genes after DNA damage. Additionally, MATR3 has also been shown to bind with pre-miR-138-2, hindering it from being further processed into the mature form [57]. Lastly, MATR3 has also been identified as an interactor of the Argonaute complexes, suggesting a possible role in miRNA-induced silencing [51]. 

Another process that MATR3 regulates is RNA splicing. Alternative splicing (AS) is a mechanism that enhances transcriptomic and proteomic diversity from a limited number of protein coding genes through the use of differing combinations of splice sites. Many reports have shown that MATR3 interacts with a variety of proteins that are known to function in splicing regulation [15,33,46,61]. Several studies have shown that the major role of MATR3 is a repressor of exon inclusion [15,16,17,19]. MATR3’s splicing function is dependent on its RRM domains, while the ZnF domains are largely dispensable [15]. Specifically, MATR3 binding is enriched in intronic pyrimidine-rich sequences around or within the repressed exons [15,17]. Moreover, several studies have demonstrated that MATR3 works together with PTBP to regulate RNA splicing, but it may also act independently [15,17,18]. 

In addition to alternative splicing, MATR3 has also been linked to non-canonical splicing. Introns of the eukaryotic genome often contains sequences that resemble splice sites or polyadenylation (poly(A)) sites, often referred to as “cryptic” sites [62]. Recognition of these cryptic sequences by the splicing machinery may lead to non-canonical splicing that may then result in the production of an aberrant transcript and/or protein. One form of non-canonical splicing is the inclusion of previously unannotated exons, also commonly referred to as cryptic exons (CEs), which often contain premature termination codons (PTCs) that can target a CE-containing transcript to nonsense-mediated decay (NMD), thereby decreasing its transcript levels. A known source of these cryptic splice or poly(A) sites are LINE elements, the most abundant transposable elements in the human genome [18,62]. Similar to its functionality in AS, MATR3 has been shown to repress the recognition of cryptic splice sites, thus repressing the inclusion of CEs [18]. This function is mediated through the binding of MATR3 in the deep intronic regions where these cryptic sites are located. Another study recently examined global MATR3-dependent cryptic splicing events (CSEs) within functional genes [16]. Conforming to previous findings [15], the RRM domains, particularly RRM2, are required for mediating the CE repression function of MATR3 [16]. 

Recent studies have also reported that MATR3 plays a role in the regulation of global protein synthesis. Although MATR3 is predominantly nuclear in localization, the presence of both a nuclear localization signal and a nuclear export signal may suggest that it may have roles in both the nucleus and the cytoplasm [63]. Indeed, a recent study identified the protein interactome of MATR3 in human induced pluripotent stem cells (hiPSCs) and revealed that MATR3 associates with members of the translational machinery [12] (Table 1). Other interaction proteomic analyses of MATR3 interactors revealed that MATR3 interacts with various ribosomal proteins [33,52,53] (Table 1). Additionally, MATR3 has been found to be partially cytoplasmic in localization in hiPSCs with a portion of cytoplasmic MATR3 (about 20%) colocalizing with members of the eIF3 complex (eIF3A and eIF3C) [12], a protein complex important for the initiation of protein synthesis [64]. Notably, depletion of MATR3 decreased the translational efficiency of pluripotency regulating genes including *NANOG*, *LIN28A*, and *SOX2* in hiPSCs [12]. Of note, an early study of MATR3 localization has also described a population of MATR3 that is detectable in the cytoplasmic, microsomal, and polysomal fractions of rat liver and Ac2F hepatoma cells [7]. 

## 4. Regulation of MATR3 Degradation

Several studies have explored how wildtype MATR3 is regulated and degraded (Figure 2). Specifically, MATR3 has been shown to be phosphorylated by protein kinase A (PKA) and then subsequently degraded in rat cerebellar granule neurons exposed to N-methyl-D-aspartate (NMDA) [65]. Furthermore, inhibition of NMDA receptors (NMDAR) or of PKA activity prevented NMDA-induced phosphorylation and the subsequent degradation of MATR3, suggesting that the NMDAR-PKA signaling is important for MATR3 degradation. Additionally, a few studies have shown that MATR3 degradation could be mediated by caspases and/or calpains. A previous study demonstrated that MATR3 is a Ca^2+^-dependent calmodulin-binding protein harboring a caspase cleavage site and that it could be cleaved by caspases-3, -5, -6, -7, -8, and -10 [66]. Another study also demonstrated that in addition to multiple caspases, MATR3 can also be cleaved by calpains 1 and 2 [67]. Complementarily, caspase inhibition has been shown to increase MATR3 levels, likely through the prevention of its caspase-mediated degradation [68]. Lastly, a recent study reported similar findings in that MATR3 abundance is regulated in an NMDAR-, Ca^2+^- and calpain-dependent manner in rat primary cortical neurons [69]. Intriguingly, this study showed that PKA inhibitor H-89 did not significantly prevent NMDA-mediated MATR3 degradation in cortical neurons [69], suggesting that MATR3 degradation may not be dependent on PKA phosphorylation, which is in contrast to the previous findings in cerebellar granule neurons [65]. This study also showed that shortly upon activation of glutamatergic signalling or after Ca^2+^ influx, calmodulin binds to MATR3 and impairs MATR3 RNA-binding capability [69]. These studies suggest that NMDAR-signalling and the resulting Ca^2+^ influx leads to the activation of calmodulin, which allows it to bind to MATR3 and displace MATR3-bound RNA, leaving MATR3 susceptible to degradation by calpains.

## 5. MATR3’s Role in the Regulation of Development and Differentiation

The protein sequence of MATR3 is highly conserved in vertebrates. It is ubiquitously expressed in various tissues, including the central nervous system, muscles, and various organs, which may suggest that MATR3 may play an important role in these organs (Figure 3). Indeed, constitutive deletion of *Matr3* in mice is perinatally lethal [70], suggesting that MATR3 may play an important role in development. Likewise, homozygous disruption of *Matr3* 3’ UTR in mice, through gene trapping (*Matr3^Gt-ex13^* mice) is early embryonically lethal [71]. Notably, *Matr3^Gt-ex13^* heterozygotes are viable but exhibit incompletely penetrant cardiac defects, suggesting a possible role of MATR3 in cardiac development. 

MATR3 has also been recently shown to be involved in the maintenance of pluripotency in hiPSCs [12]. Stable depletion of MATR3 resulted in reduced growth and colony formation capability in suboptimal single-cell growth conditions. The downregulation of MATR3 also led to a decrease in the levels of pluripotency factors, including OCT4, NANOG, KLF4, and YTHDF1. Notably, overexpression of MATR3 in MATR3-depleted hiPSCs rescued the reduced growth phenotype and the reduction in pluripotency factors suggesting that these phenotypes are indeed mediated by the loss of MATR3. Furthermore, neurons differentiated from MATR3-depleted hiPSCs displayed a reduction in neurite length and arborization, suggesting that in addition to maintaining stemness, MATR3 also plays a role in terminal neuronal maturation.

In addition to hiPSCs, MATR3 has also been reported to affect the proliferation and/or differentiation of other stem cells or precursor cells. Knockdown of MATR3 in neural stem cells (NSCs) decreased cell proliferation and neurosphere formation and also induced the differentiation of these stem cells, as observed through the extension of neurites [20]. Additionally, in utero knockdown of MATR3 in mouse hippocampal regions where NSCs reside resulted in the failure of MATR3-depleted cells to migrate to the cortical plate and differentiate into neurons, resulting in the disorganization of this region. Similarly, depletion of MATR3 in murine primary myoblasts also decreased proliferation and impaired proper differentiation [21]. Depletion of MATR3 in the mouse myoblast cell line C2C12 similarly led to an impairment in muscle differentiation [11]. Lastly, knockout of *Matr3* in murine erythroid cells and mouse embryonic stem cells resulted in phenotypes indicative of a state of accelerated differentiation [10]. 

MATR3 was also shown to be important for cell survival and growth in other cells. Depletion of MATR3 reduced the proliferation and altered the morphology of endothelial cells [23]. Furthermore, prolonged depletion of MATR3 led to a decrease in cell viability and an increase in cell death. In primary cortical neurons, depletion of MATR3 is similarly toxic [22]. Lastly, a recent study has demonstrated that MATR3 regulates mitotic spindle dynamics and cell proliferation by controlling the alternative splicing of *CDC14B* in colorectal cancer cells [19].

## 6. MATR3 in the Context of Disease

### 6.1. Matrin-3 Mutations in Amyotrophic Lateral Sclerosis

Mutations in *MATR3* have been linked to amyotrophic lateral sclerosis (ALS), a devastating adult-onset neurodegenerative disease that is characterized by the loss of motor neurons, leading to muscle atrophy, paralysis, and death [24,72] (Table 2). Intriguingly, ALS-associated missense mutations in *MATR3* cluster outside of its functional domains [24,73,74,75,76,77,78], particularly in the intrinsically disordered regions (IDRs) (Figure 4). Of these *MATR3* mutations, the most common mutation is the Serine-85 to Cysteine (S85C) mutation located in the N-terminal IDR of MATR3. The S85C mutation was initially linked to another adult-onset disorder, namely vocal cord and pharyngeal weakness with distal myopathy (VCPDM) [79,80], but was later reclassified as slowly progressive ALS due to the presence of neurogenic features involving motor neuron symptoms [24]. Notably, MATR3 S85C has also been associated to VCPDM in several additional families with or without similar neurogenic features [81,82,83,84,85,86,87], suggesting that this mutation may cause a spectrum of phenotypes ranging from pure myopathy to motor neuron disease. Similarly, mice that harbor the S85C mutation in the mouse *Matr3* gene also show robust phenotypes, including motor function defects, muscle atrophy, and early death [70], suggesting that the S85C mutation is a genetic determinant of myopathy and ALS. Conversely, the Phe115Cys (F115C) mutation in *MATR3* has been initially linked to familial ALS [24], but resequencing on the same ALS family revealed an intronic mutation in another ALS-linked gene *KIF5A* [88], suggesting that the F115C mutation may not be causative of ALS. This finding was supported by the lack of pathogenicity of the F115C mutation in a knock-in mouse model [89]. The Pro154Ser (P154S) mutation is another ALS-linked mutation in *MATR3*, initially found in a sporadic ALS patient [24]. Previous studies in various cell lines have shown that expression of MATR3 P154S leads to alterations in nuclear mRNA export [53], impairment in MATR3 condensate dynamics in yeast [90], and increased toxicity compared to MATR3 WT in rat primary cortical neurons [22]. Although these studies suggest that the P154S mutation may impact MATR3 function and cause toxicity, the MATR3 P154S knock-in mice did not display any overt phenotypes, suggesting that the P154S mutation may not induce pathogenicity [91]. In addition, the Thr622Ala (T622A) mutation was another mutation in *MATR3* that was identified in familial ALS [24]. Similar to the P154S mutant, expression of MATR3 T622A resulted in an impairment in MATR3 condensate dynamics in yeast [90] and increased toxicity relative to MATR3 WT [22]. Notably, although global mRNA export is not affected by the expression of MATR3 T622A, defects in the mRNA export of other ALS-linked genes *TDP43* and *FUS* were observed in MATR3 T622A-expressing cells [53].

**Figure 4 cells-13-00980-f004:**
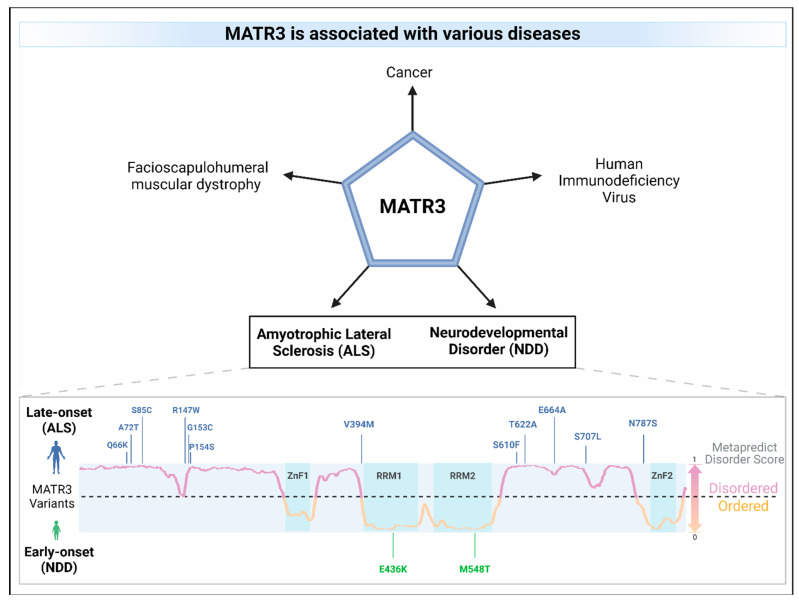
MATR3 is associated with various diseases. Missense mutations in *MATR3* have been linked to amyotrophic lateral sclerosis (ALS; mutations in blue) as well as to neurodevelopmental disorders (NDD; mutations in green). ALS-associated mutations are located in the intrinsically disordered regions of MATR3, as predicted using the Metapredict v2 webserver (metapredict.net) [92] while missense mutations in the ordered regions were identified in NDDs. A residue having a Metapredict disorder score above 0.5 indicates that it is likely to be in an intrinsically disordered region (lilac), whereas a score below 0.5 indicates that it is likely to be in an ordered region (orange). The domains of MATR3 were indicated (zinc finger, ZnF; RNA recognition motif, RRM) and colored in turquoise.

**Table 2 cells-13-00980-t002:** Combined annotation-dependent depletion (CADD) score of disease-associated missense mutations in *MATR3*.

Associated Disease *	HGVS.c (NM_199189.3)	HGVS.p (NP_954659.1)	CADD GRCh37-v1.7		CADD GRCh38-v1.7		Ref
			Position (GRCh37)	CADD Score (PHRED)	Position (GRCh38)	CADD Score (PHRED)	
ALS	c.151C>T	p.(Arg51Cys)	138643255	35	139307566	34	[72]
ALS	c.182G>A	p.(Ser61Asn)	138643286	33	139307597	33	[72]
ALS	c.196C>A	p.(Gln66Lys)	138643300	34	139307611	29.9	[77]
ALS	c.214G>A	p.(Ala72Thr)	138643318	31	139307629	29.4	[73]
ALS	c.254C>G	p.(Ser85Cys)	138643358	35	139307669	35	[24]
ALS	c.296C>G	p.(Ser99Cys)	138643400	35	139307711	35	[72]
ALS	c.393C>A	p.(Asp131Glu)	138643497	27	139307808	32	[72]
ALS	c.439A>T	p.(Arg147Trp)	138643543	35	139307854	35	[74]
ALS	c.457G>T	p.(Gly153Cys)	138643561	35	139307872	35	[77]
ALS	c.460C>T	p.(Pro154Ser)	138643564	27.8	139307875	25.1	[24]
ALS	c.545G>A	p.(Arg182Lys)	138643649	34	139307960	34	[72]
ALS	c.561T>G	p.(Asp187Glu)	138643665	28.6	139307976	32	[72]
ALS	c.883A>G	p.(Ile295Val)	138643987	34	139308298	34	[72]
ALS	c.926A>G	p.(His309Arg)	138650377	28.2	139314688	28.8	[72]
ALS	c.949C>T	p.(Arg317Cys)	138650400	34	139314711	34	[72]
ALS	c.998A>G	p.(Asn333Ser)	138651409	19.48	139315720	18.05	[72]
ALS	c.1102C>A	p.(Pro368Thr)	138651850	24.7	139316161	24.2	[72]
ALS	c.1175G>T	p.(Gly392Val)	138652787	25.2	139317098	25.6	[72]
ALS	c.1180G>A	p.(Val394Met)	138652792	22.8	139317103	22.5	[75]
ALS	c.1282C>A	p.(His428Asn)	138653384	28.2	139317695	25.1	[72]
NDD	c.1306G>A	p.(Glu436Lys)	138653408	28.7	139317719	31	[25]
NDD	c.1643T>C	p.(Met548Thr)	138657627	25.5	139321938	25.7	[16]
ALS	c.1786T>A	p.(Ser596Thr)	138658294	20.2	139322605	18.97	[72]
ALS	c.1829C>T	p.(Ser610Phe)	138658337	28.6	139322648	27	[76]
ALS	c.1837G>C	p.(Asp613His)	138658345	26.6	139322656	25.8	[72]
ALS	c.1864A>G	p.(Thr622Ala)	138658372	0.189	139322683	1.475	[24]
ALS	c.1867G>A	p.(Glu623Lys)	138658375	22.3	139322686	22.8	[72]
ALS	c.1879C>G	p.(Gln627Glu)	138658387	20.6	139322698	18.76	[72]
ALS	c.1921G>C	p.(Asp641His)	138658429	23.3	139322740	23	[72]
ALS	c.1948A>C	p.(Met650Leu)	138658456	18.49	139322767	17.49	[72]
ALS	c.1991A>C	p.(Glu664Ala)	138658499	24.3	139322810	23.6	[77]
ALS	c.2062G>T	p.(Ala688Ser)	138658570	7.085	139322881	8.125	[72]
ALS	c.2075A>G	p.(Lys692Arg)	138658583	22.3	139322894	23.4	[72]
ALS	c.2120C>T	p.(Ser707Leu)	138658628	22.2	139322939	18.97	[77]
ALS	c.2135A>G	p.(Lys712Arg)	138658643	22.1	139322954	24	[72]
ALS	c.2203A>G	p.(Ile735Val)	138661183	18.45	139325494	18.66	[72]
ALS	c.2219A>G	p.(Asn740Ser)	138661199	18.61	139325510	16.56	[72]
ALS	c.2234C>T	p.(Ala745Val)	138661214	21.8	139325525	21	[72]
ALS	c.2251G>A	p.(Ala751Thr)	138661231	19.59	139325542	17.87	[72]
ALS	c.2275A>G	p.(Ser759Gly)	138661255	22.3	139325566	21.8	[72]
ALS	c.2360A>G	p.(Asn787Ser)	138661340	21.3	139325651	20.7	[77]
ALS	c.2504A>G	p.(Asn835Ser)	138665044	17.22	139329355	17.24	[72]

Missense mutations in red were obtained from the previous literature, whereas missense mutations in black were obtained from the Project MinE data browser (AF_controls_ = 0). CADD scores, which indicate the predicted deleteriousness of each mutation, were obtained using CADD v1.7 [93]. * Amyotrophic lateral sclerosis, ALS; neurodevelopmental diseases, NDD.

As accumulating genetic studies corroborate that the S85C mutation in *MATR3* is pathogenic, various animal models were used to mimic the genetic condition and address the mechanism by which the S85C mutation causes the disease. Several mouse models of MATR3 S85C has been generated. Constitutive global overexpression of MATR3 S85C in mice resulted in myopathic phenotypes, spinal motor neuron degeneration, increased glial markers suggestive of neuroinflammation in the spinal cord, impairment in weight gain, motor function defects, and decreased lifespan [94]. However, the phenotypic comparisons were made with non-transgenic control and not with transgenic MATR3 WT mice, which complicates the interpretation of the resulting phenotypes as either stemming from the S85C mutation or an artifact of MATR3 overexpression. To counter this, a knock-in mouse model that harbors the S85C mutation in the endogenous mouse *Matr3* locus was generated [70]. Homozygous MATR3 S85C knock-in mice (*Matr3^S85C/S85C^*) recapitulate key features of early-stage ALS, including an impairment in weight gain and a progressive and age-dependent deficit in motor function, including impairment in motor coordination and gait, as well as muscle weakness. Neuropathological analyses at the disease end stage revealed a striking loss of Purkinje cells in the cerebellum and defects in the neuromuscular junction (NMJ). Interestingly, these ALS mice show a profound loss of MATR3 staining in a subset of neurons affected in ALS, such as in Purkinje cells and alpha motor neurons, but not in gamma motor neurons, which are typically spared in ALS. The relevance of this presymptomatic MATR3 loss in these neurons is still unclear and requires further study; however, the depletion of MATR3 has been previously shown to be toxic to cortical neurons in vitro [22], suggesting that MATR3 loss may contribute to Purkinje cell loss and motor function defects. Further characterization of MATR3 S85C knock-in mice extended the population of neurons that display MATR3 loss to include the alpha motor neurons and interneurons in the cervical and thoracic spinal cord, and a subset of upper motor neurons and hippocampal CA1 neurons in the brain [95], without being associated with cell body loss. These results suggest that the S85C mutation affects selective neuronal populations in other regions of the brain in addition to motor-controlling neurons, which requires further investigation.

Additional studies support the hypothesis that the S85C mutation leads to partial loss of MATR3 function. Characterization of MATR3 S85C from multiple models consistently show that the S85C mutant is less soluble compared to wild type (WT) MATR3 [16,22,24,80,96,97], suggesting that the S85C mutation reduces the levels of functional MATR3. However, a concomitant increase in the insoluble levels of S85C was observed compared to the wildtype MATR3, suggesting that the S85C mutation may change the biochemical properties of the MATR3 protein, which may cause toxicity. Additionally, the S85C mutation has been found to impair the CE repression capability of MATR3, further supporting the idea that the S85C mutation is a partial loss-of-function mutation [16].

Other recent studies have also provided evidence on how MATR3 S85C may cause disease (Figure 5). Indeed, studies have shown that MATR3 S85C causes toxicity in various model systems, including human and mouse cell lines, primary neurons, yeast, and *Drosophila* [9,22,90,96,97,98]. In a motor neuron-like NSC-34 cell line, the S85C mutation was shown to alter the protein–protein interactions and/or colocalization of MATR3 with members of the TREX complex, leading to defects in nuclear mRNA export [53]. However, another study conducted on HEK293 cells reported that the protein interactome of MATR3 S85C is similar to that of the WT protein and did not identify MATR3 interaction with TREX complex members, suggesting cell-type specific MATR3 interactions [52]. In addition, MATR3 S85C was recently shown to be resistant to degradation by calpain-1, an enzyme previously reported to degrade MATR3 WT [67,69], suggesting a possible gain-of-function mechanism for this mutation [69]. Moreover, it has been shown that MATR3 S85C was not sufficient to rescue the redistribution of H3K27me3-modified chromatin upon depletion of MATR3 in mouse neuroblast cells and that the S85C mutant protein was less dynamic compared to the wildtype MATR3 protein [9], suggesting that dysregulation of chromatin organization and defects in protein dynamics may contribute to the mutation’s disease-causing potential. In a yeast model, the S85C mutation was shown to also impair condensate dynamics [90]. Intriguingly, it was previously shown that a fragment of wildtype MATR3 that contains the N-terminal IDR can phase-separate when targeted to the nucleus [99]. Conforming to the previous two findings, introduction of the S85C mutation into this phase-separating fragment similarly impaired droplet formation [99]. These studies suggest that the S85C mutation seems to affect the phase separation capability of MATR3, but how this impairment in phase separation mediates toxicity is not yet known. Stress granule formation upon treatment of the stressor was also shown to be impaired in S85C fibroblasts, evidenced by a reduction in the formation of stress granules (SGs) within cells [98]. Furthermore, findings from two independent groups have shown that expressing MATR3 in the motor neurons or muscles of flies led to an impairment in motor function and shortened lifespan with mutant MATR3 S85C being more toxic compared to MATR3 WT [96,97]. Candidate genetic screens from both groups identified that knockdown of axonal transport genes enhances toxicity [97], while knockdown of the *Drosophila* homolog of *HNRNPM* suppressed the toxicity of MATR3 S85C but not MATR3 WT [96]. How the S85C mutation affects MATR3 function and causes toxicity is beginning to be unraveled, and further studies will help accelerate our understanding of the disease and to develop treatments for patients.

### 6.2. Matrin-3 Mutations in Neurodevelopmental Diseases

Missense mutations in *MATR3* have also been linked to neurodevelopmental diseases (Figure 4 and Figure 6). Intriguingly, two reported neurodevelopmental disease-associated *MATR3* variants contain mutations in the ordered domains of MATR3, particularly in the RRM domains. The E436K mutation, located in the RRM1 domain of MATR3, has been reported to lead to haploinsufficiency with reduced MATR3 levels [25]. This patient displayed developmental disability, muscular hypotonia, and early-onset neurodegeneration. Another patient with a MATR3 M548T mutation, located in the RRM2 domain, was recently reported [16]. Similar to the patient with the E436K mutation, the patient with the M548T mutation was reported to have motor phenotypes and intellectual disability. Characterization of the MATR3 M548T variant revealed that it is deficient in RNA binding and CE repression capability when compared to MATR3 WT, suggesting that this mutation impairs RRM2 function. However, it is still unknown whether the E436K variant in the RRM1 is similarly deficient in RNA binding and CE repression capabilities. Comparing the phenotypic trajectory and age of onset of the two neurodevelopmental disease-associated MATR3 variants to the ALS-associated MATR3 S85C variant, it seems to suggest that mutations in the ordered domains of MATR3 may be linked to more severe phenotypes earlier in life. How these disease-associated variants in MATR3 lead to different disease onset and progression is still currently unclear but, nonetheless, it is an interesting conundrum that requires further research.

### 6.3. Matrin-3 Dysregulation in Other Diseases

Dysregulation in MATR3 expression has been linked to other diseases, including cancer (Figure 4). Elevated levels of MATR3 are associated with lower overall survival, advanced clinical stage, and/or higher tumor grade in patients with neuroblastoma [100] or hepatocellular carcinoma (HCC) [26]. Downregulation of MATR3 in HCC cell lines resulted in decreased cell proliferation and increased cell cycle arrest, suggesting that MATR3 promotes cell proliferation in these cell lines. Similarly, knockdown of MATR3 decreased cell proliferation and/or increased apoptosis in oral squamous cell carcinoma cells [27], malignant melanoma cells [28], and colorectal cancer cells [19]. Moreover, depletion of MATR3 in a xenograft mouse model significantly reduced tumoral volume and size compared to control, supporting an oncogenic role for MATR3 [26,28]. On the other hand, there are also a few studies reporting an opposite role for MATR3, suggesting its tumor-suppressor-like function. In triple-negative breast cancer (TNBC), high levels of MATR3 promoted apoptosis and inhibited epithelial-mesenchymal transition, migration, and invasion of the cancer cells [29]. Additionally, high MATR3 levels are associated with better prognosis in breast cancer patients. Similarly, high MATR3 protein expression was also associated with higher overall survival in patients with non-small cell lung cancer [30] or clear-cell renal cell carcinoma [31]. Taken together, these studies demonstrate that MATR3 dysregulation is associated with cancer, but it may play a different role depending on the cancer type. More studies are required to examine why MATR3 plays a differential role in these cancer types as well as to pinpoint the exact mechanisms as to how MATR3 contributes to cancer disease progression. 

Beyond cancer, MATR3 has also been reported to have ties to other diseases, such as facioscapulohumeral muscular dystrophy (FSHD). FSHD is a neuromuscular disorder caused by the de-repression of the transcription factor *DUX4*, whose target genes are known to be toxic to muscle cells [55]. In FSHD muscle cells, MATR3 has been shown to bind to the DUX4 DNA-binding domain and inhibit DUX4-mediated gene expression, thereby reducing apoptosis and improving myogenic defects [55]. Intriguingly, these effects seem to be mediated by the N-terminal IDR of MATR3 as it is both necessary and sufficient for the inhibition of the DUX4-mediated phenotypes. 

Lastly, the role of MATR3 has also been studied in the regulation of human immunodeficiency virus (HIV) gene expression. Several studies have demonstrated that MATR3 is a cofactor and modulator of Rev, an HIV-1 encoded protein that functions in the nuclear export of viral mRNA [101,102,103]. Furthermore, depletion of MATR3 inhibited HIV-1 replication [103]. Conversely, overexpression of MATR3 augmented HIV-1 replication. However, MATR3’s effect on HIV-1 viral replication may be mediated by its ability to modulate the inhibition of HIV-1 through ZAP, a host antiviral protein, through its interaction with ZAP and the ZAP degradation complex [54]. 

## 7. Conclusions

Since the discovery of MATR3 as a nuclear matrix protein about 30 years ago, our understanding of its roles has greatly expanded. We now know that MATR3 is not merely a scaffolding protein; it has also been found to play a pleiotropic role at multiple levels during gene expression, contributing to chromatin organization, transcription, RNA metabolism, and translation by collaborating with various protein partners. Additionally, accumulating evidence also revealed that MATR3 may play a critical role in the DNA damage response. In terms of its cellular function, MATR3 plays an important role in cell proliferation, differentiation, and survival. More studies are required to reveal how MATR3 works together with its interactors to regulate gene expression in a specific cellular context. After the identification of disease-associated mutations, MATR3 has become a more attractive target for further study. We now have some clues as to how the ALS-linked S85C mutation alters MATR3 function and causes disease; however, additional studies are required to understand how these disease-associated mutations alter MATR3’s structure, properties, and function, and how this mutant protein leads to cellular dysfunction and cause disease. Recent studies have also revealed MATR3’s involvement in other diseases, including cancer, further underscoring the importance in studying the role of MATR3 in various cellular and disease contexts. Determining the role of MATR3 would help identify potential therapeutic targets for various diseases, from neurodegeneration to cancer, and enable development of therapeutic strategies.

## Figures and Tables

**Figure 1 cells-13-00980-f001:**
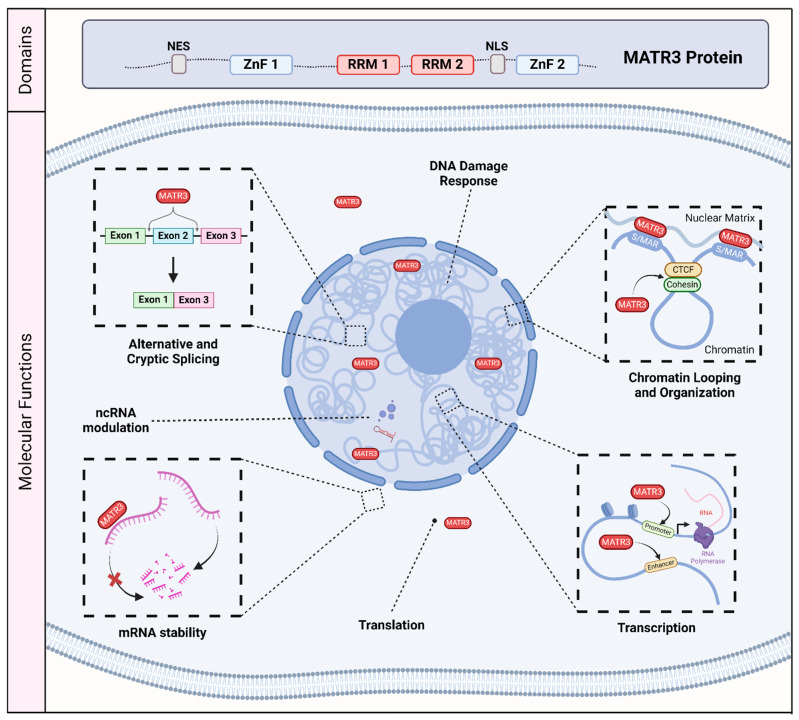
The molecular and biological roles of MATR3. MATR3 contains a nuclear export signal (NES), a nuclear localization signal (NLS), as well as two zinc finger (ZnF) domains and two RNA recognition motifs (RRM). As a nucleic acid binding protein, MATR3 plays a role in various process associated with DNA and RNA, including the DNA damage response, chromatin looping and organization, transcription, translation, mRNA stability, modulation of noncoding RNA (ncRNA), and RNA splicing. Scaffold/matrix attachment region, S/MAR; CCCTC-binding factor, CTCF.

**Figure 2 cells-13-00980-f002:**
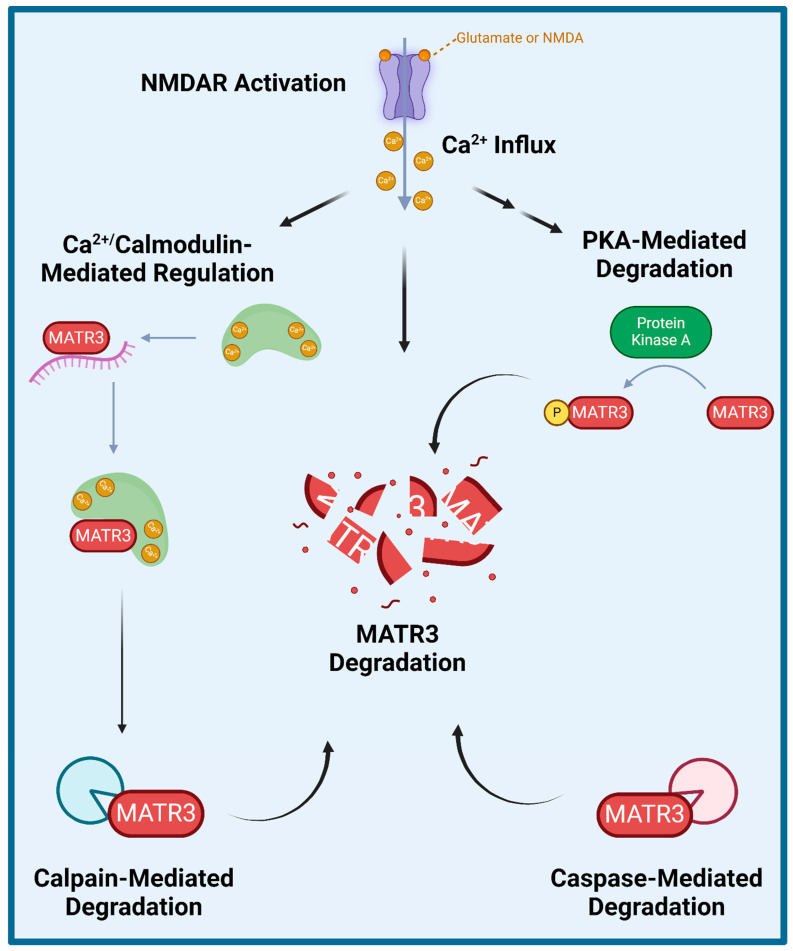
Proposed mechanisms of MATR3 degradation. Several studies have shown that MATR3 can be degraded through a variety of mechanisms including enzymatic cleavage mediated by calpains and a variety of caspases, and following activation of N-methyl D-aspartate (NMDA) receptor (NMDAR) or phosphorylation catalyzed by protein kinase A (PKA).

**Figure 3 cells-13-00980-f003:**
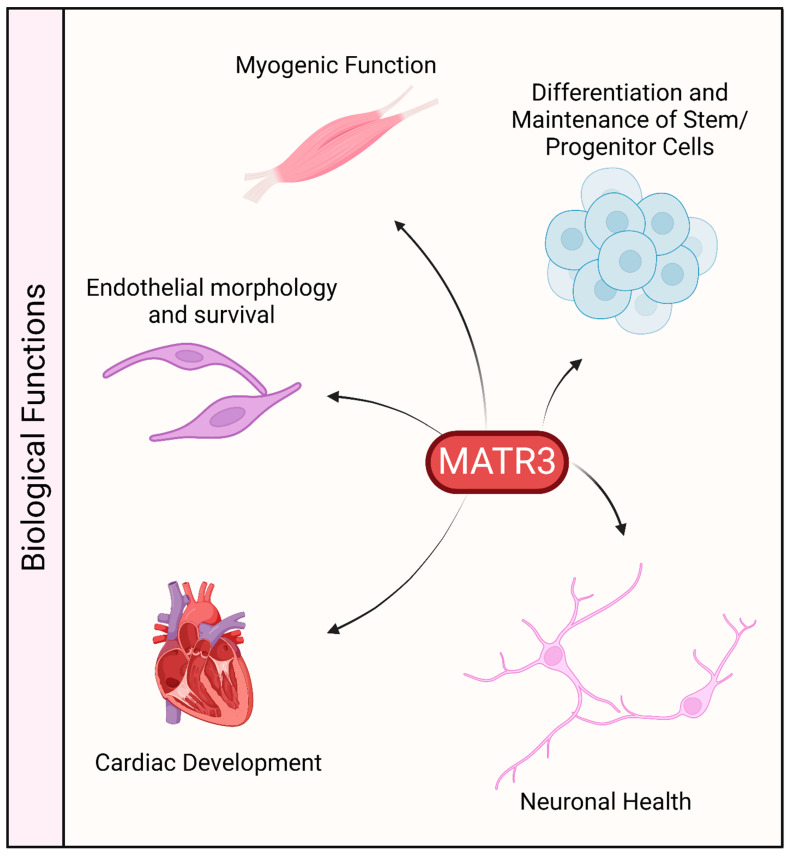
Biological functions associated with MATR3. MATR3 has been shown to play a role in maintaining or modulating the function and/or development of a variety of cell types.

**Figure 5 cells-13-00980-f005:**
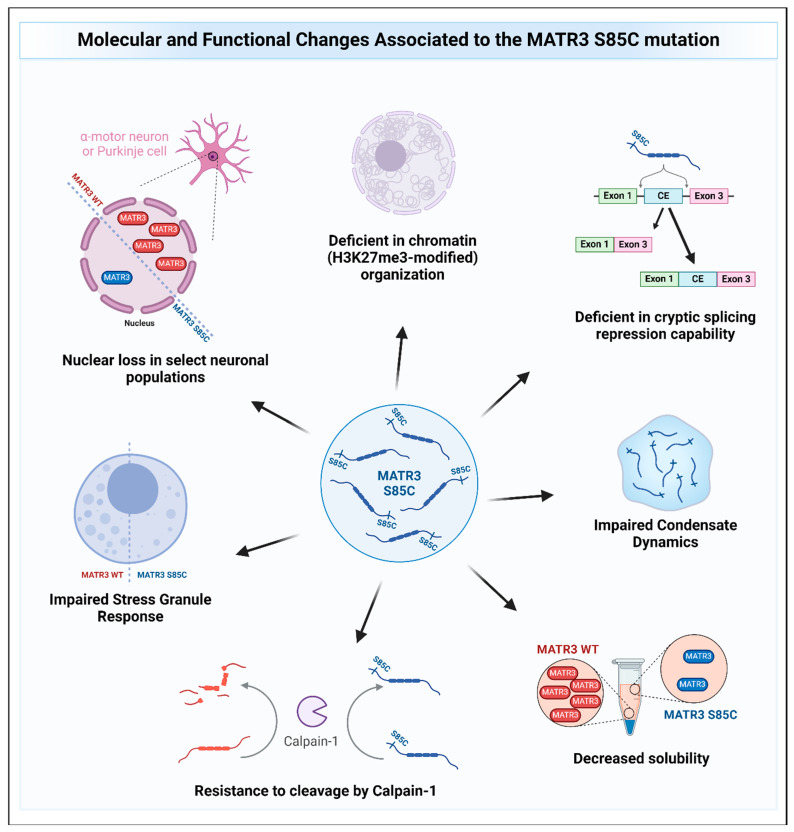
The molecular and functional changes associated with the MATR3 S85C mutation. Pathological changes to MATR3 function and properties due to the missense S85C mutation include, but are not limited to, deficiencies in its normal function and impairment of its biochemical properties.

**Figure 6 cells-13-00980-f006:**
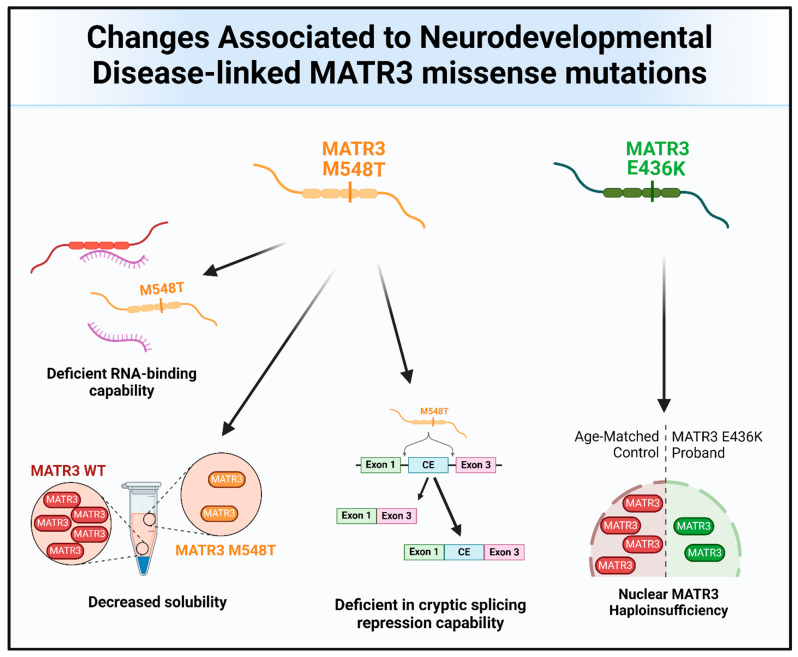
The molecular and functional changes associated with the MATR3 neurodevelopmental disease-associated mutations. Pathological changes to MATR3 function and properties due to the neurodevelopmental disease-associated missense mutation include, but are not limited to, deficiencies in its normal function and impairment of its biochemical properties.

## Data Availability

Not applicable.

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
