# Peer review of "MATR3’s Role beyond the Nuclear Matrix: From Gene Regulation to Its Implications in Amyotrophic Lateral Sclerosis and Other Diseases"

_cells, 2024, doi:10.3390/cells13110980_

Round 1

Reviewer 1 Report

Comments and Suggestions for Authors

This is a very detailed review! The authors collected many references to write this logical and easy-to-read review, which is valuable for MATR3 research. I appreciate their work. I have only two minor suggestions, which I hope can further enhance this manuscript.

1. Title: I suggest using an affirmative Sentence, like" The physiological roles of MATR3 and functional alteration by a single amino acid mutation in various diseases".

2. Add a Table listing MATR3-associated DNA, RNA, and proteins described in the text.

Author Response

Reviewer 1

This is a very detailed review! The authors collected many references to write this logical and easy-to-read review, which is valuable for MATR3 research. I appreciate their work. I have only two minor suggestions, which I hope can further enhance this manuscript.

1. Title: I suggest using an affirmative Sentence, like" The physiological roles of MATR3 and functional alteration by a single amino acid mutation in various diseases".

Response) Thank you for your positive feedback and appreciation of our work. We are delighted that you found our review detailed, logical, and easy to read. We also appreciate your suggestion regarding the title of our review article. As such, we have decided to rename our article as “MATR3’s role beyond the nuclear matrix: From gene regulation to dysregulation in disease”.

2. Add a Table listing MATR3-associated DNA, RNA, and proteins described in the text.

Response) Thank you for your suggestion to add a table listing the MATR3 interactors described in the text. We agree that such a table would enhance the accessibility of the information presented in our manuscript. In response, we have created a new table (Table 1) listing the interactors of MATR3 that we have touched upon in the review article with the accompanying reference.

Reviewer 2 Report

Comments and Suggestions for Authors

My suggestions:

1. For MATR3 mutations, I would add a table, that summarizes the involvement of mutations in diseases, their CADD score, and location.

2.. Is there a PDB structure for MATR3 in the Protein Data Bank?  The authors may add a PDB structure to the manuscript made by Phyre2 or AlpaFold Colab.

3. A few more figures may improve the manuscript further. For example, adding one in the chapter "Regulation of MATR3 degradation" would be nice.

4. Were there any additional MATR3 mutations discovered in ALS and NDD cases besides the ones the authors described? 

5. I would make a separate figure on the role of mutations in neurodevelopmental diseases.

Author Response

1. For MATR3 mutations, I would add a table, that summarizes the involvement of mutations in diseases, their CADD score, and location.

Response) We thank the reviewer for their suggestion. We agree that such a table would enhance the quality of our review article. As such, we have now included a new table (Table 2) in the revised manuscript that provides additional information on the mutations listed on Figure 4, as well as on new ALS-associated mutations obtained from the Project MinE database.

2.. Is there a PDB structure for MATR3 in the Protein Data Bank?  The authors may add a PDB structure to the manuscript made by Phyre2 or AlpaFold Colab.

Response) Thank you for your suggestion regarding the inclusion of a PDB structure for MATR3 in our manuscript. Currently, only the structures for the two RNA recognition motifs (RRMs) of MATR3 are experimentally verified and deposited on the Protein Data Bank (PDB ID: 7FBR for RRM1 and 7FBV for RRM2), and therefore added this reference (He et al. Biomolecular NMR Assignments, 2021) to our manuscript. While the AlphaFold Protein Structure Database includes a predicted structure for MATR3, after careful consideration, we have decided not to add the predicted structure to our manuscript as we believe that the intrinsically disordered property of MATR3 is better reflected by a per-residue disorder plot as we have shown in Figure 4.

3. A few more figures may improve the manuscript further. For example, adding one in the chapter "Regulation of MATR3 degradation" would be nice.

Response) Thank you for your suggestion to include additional figures in our manuscript. In response to your recommendation, we have now added additional figures illustrating the discussed mechanisms of MATR3 degradation (Figure 2) as well as the changes associated with MATR3 mutations associated with neurodevelopmental diseases (Figure 6).

4. Were there any additional MATR3 mutations discovered in ALS and NDD cases besides the ones the authors described? 

Response) For this review article, we have only reported on ALS-associated or NDD-associated MATR3 mutations identified in specific patients or cohorts. However, this did not include any mutations discovered in larger or ongoing genetic studies such as Project MinE. To better reflect the breadth of disease-associated mutations in MATR3, we have now included other ALS-associated missense mutations in MATR3 identified by Project MinE with the accompanying CADD score and location for each mutation (Table 2). Thank you for bringing attention to this important aspect.

5. I would make a separate figure on the role of mutations in neurodevelopmental diseases.

Response) We thank you for your suggestion. We have now added a new figure (Figure 6) illustrating the changes associated to the MATR3 E436K and M548T variants.

Reviewer 3 Report

Comments and Suggestions for Authors

In the manuscript “What do we know so far about the physiological roles of MATR3 and how is MATR3 function altered in various diseases?” by Santos and Park, the authors went beyond the regular scaffolding role of MATR3 as an inner nuclear matrix protein. They discussed the roles of MATR3 in multiple DNA- and RNA-associated processes involving chromatin organization, DNA transcription and repair, and RNA splicing. The authors also reviewed disease-associated MATR3 mutations and focused on their links to Amyotrophic Lateral Sclerosis and Neurodevelopmental Disorder. In summary, this review provides a unique perspective of MATR3 functions in various cell types related to embryonic development, terminal neuronal maturation, and cell proliferation.   

Some minor suggestions are:

1.    Provide references for some sentences in the introduction. For example, page 1 line 37, “However, accumulating studies since its discovery have demonstrated its role in various molecular and cellular processes involving DNA and RNA, including chromatin organization, DNA transcription and repair, and RNA splicing.” The authors can cite review papers if too many primary research papers are involved. 

2.    The authors provided a detailed summary of various defects of an important MATR3 mutant, S85C. However, the authors might add a simple paragraph summarizing the defects of other disease-relevant MATR3 variants listed in Figure 2, other than E436K and M548T.  

Author Response

In the manuscript “What do we know so far about the physiological roles of MATR3 and how is MATR3 function altered in various diseases?” by Santos and Park, the authors went beyond the regular scaffolding role of MATR3 as an inner nuclear matrix protein. They discussed the roles of MATR3 in multiple DNA- and RNA-associated processes involving chromatin organization, DNA transcription and repair, and RNA splicing. The authors also reviewed disease-associated MATR3 mutations and focused on their links to Amyotrophic Lateral Sclerosis and Neurodevelopmental Disorder. In summary, this review provides a unique perspective of MATR3 functions in various cell types related to embryonic development, terminal neuronal maturation, and cell proliferation.   

Some minor suggestions are:

1. Provide references for some sentences in the introduction. For example, page 1 line 37, “However, accumulating studies since its discovery have demonstrated its role in various molecular and cellular processes involving DNA and RNA, including chromatin organization, DNA transcription and repair, and RNA splicing.” The authors can cite review papers if too many primary research papers are involved. 

Response) Thank you for your valuable feedback and suggestion to provide references to support our statements in the introduction. We appreciate your attention to detail and have now included appropriate citations to primary articles to support our statement on page 1, line 37 as well as our statements on lines 40, 45, and 46.

2. The authors provided a detailed summary of various defects of an important MATR3 mutant, S85C. However, the authors might add a simple paragraph summarizing the defects of other disease-relevant MATR3 variants listed in Figure 2, other than E436K and M548T.  

Response) Thank you for your insightful suggestion. Although much of the work on MATR3 mutations is focused on the understanding of the S85C mutation, there are a handful of studies that have examined the effect of other mutations, namely the P154S and T622A mutations. We agree that expanding on the defects associated with these two MATR3 variants would be a helpful addition to our manuscript. Below is a new paragraph expanding on the defects of these additional MATR3 variants.

“Previous studies in various cell lines have shown that expression of MATR3 P154S leads to alterations in nuclear mRNA export [54], impairment in MATR3 condensate dynamics in yeast [89], and increased toxicity compared to MATR3 WT in rat primary cortical neurons [22]. Although these studies suggest that the P154S mutation may impact MATR3 function and cause toxicity, the MATR3 P154S knock-in mice did not display any overt phenotypes, suggesting that the P154S mutation may not induce pathogenicity [90]. In addition, the Thr622Ala (T622A) mutation was another mutation in MATR3 that was identified in familial ALS [24]. Similar to the P154S mutant, expression of MATR3 Thr622Ala (T622A) resulted in an impairment in MATR3 condensate dynamics in yeast [89] and increased toxicity relative to MATR3 WT [22]. Notably, although global mRNA export is not affected by the expression of MATR3 T622A, defects in the mRNA export of other ALS-linked genes TDP43 and FUS were observed in MATR3 T622A-expressing cells [54].”

Round 2

Reviewer 2 Report

Comments and Suggestions for Authors

The authors fulfilled my suggestions. Thank you.